# Rapid, Precise, and Clinically Relevant Quantification of Urinary Albumin and Creatinine Using a NanoDrop UV/Vis Spectrophotometer

**DOI:** 10.3390/s25113307

**Published:** 2025-05-24

**Authors:** Keith E. Dias, Karly C. Sourris, Jay C. Jha, Karin Jandeleit-Dahm, Bayden R. Wood

**Affiliations:** 1Monash Biospectroscopy Group, School of Chemistry, Monash University, Clayton, VIC 3800, Australia; keith.dias@monash.edu (K.E.D.); karin.jandeleit-dahm@monash.edu (K.J.-D.); 2Department of Diabetes, School of Translational Medicine, Monash University, Melbourne, VIC 3004, Australia; karly.sourris@monash.edu (K.C.S.); jay.jha@monash.edu (J.C.J.); 3German Diabetes Centre, Institute for Clinical Diabetology, Leibniz Centre for Diabetes Research at Heinrich Heine University, 40225 Dusseldorf, Germany

**Keywords:** urinary albumin, UV/Vis spectroscopy, diabetic kidney disease

## Abstract

Albuminuria is a sensitive biomarker of kidney dysfunction, and the albumin/creatinine ratio (ACR) is an essential measure for monitoring diabetic kidney disease (DKD). Abnormal levels can indicate a propensity for progressive renal failure and other complications such as cardiovascular diseases. This study employed UV/Visible spectroscopy to analyze aqueous urine samples spiked with bovine serum albumin (BSA) and creatinine at clinically relevant concentrations (0–30 mg/L for albumin and 600–1800 mg/L for creatinine) using a multivariate method. UV/Visible spectra of co-spiked samples recorded in triplicate revealed distinct bands at 229 nm and 249 nm, corresponding to BSA and creatinine, respectively, alongside other amino acid bands. Partial Least Squares Regression (PLS-R) analysis for BSA yielded a Root Mean Square Error of Calibration (RMSEC) and Cross-Validation (RMSECV) values of 66.93 and 73.92 mg/L, respectively. For creatinine, RMSEC and RMSECV values were 244.32 and 275.65 mg/L, respectively. Prediction models for both BSA and creatinine compared to ELISA demonstrated a robust performance with R^2^_PRED_ values of 0.96 and 0.95, respectively, indicating strong model reliability. The Limit of Detection (LOD) for co-spiked samples was 19.82 mg/L for BSA and 58.43 mg/L for creatinine. The significance of the achieved Limit of Detection (LOD) lies in its ability to measure concentrations well below the normal physiological ranges of 0–30 mg/L for albumin and 600–1800 mg/L for creatinine. These results demonstrate the proof of concept of applying an UV/Visible-spectroscopy-based method as a rapid, cost-effective point-of-care (PoC) tool for ACR measurements, offering promising applications in the early diagnosis, monitoring, and prognosis of diabetic kidney disease and associated cardiovascular complications. The next stage will involve a pilot trial to evaluate the technology’s potential using clinical patients.

## 1. Introduction

Diabetes is a condition characterized by excessive blood sugar (glucose) levels. Over time, this excess blood sugar can damage the nephrons, the functional units within the kidneys, leading to the development of diabetic kidney disease (DKD) [1]. DKD is typically marked by a decline in the kidney’s filtration function, known as the estimated glomerular filtration rate (eGFR), and/or an increase in the excretion of urinary albumin. As DKD progresses, there is a risk of developing a significant reduction in renal function, culminating in end-stage renal disease (ESRD). ESRD requires dialysis or kidney transplantation and is associated with increased mortality [1]. Indeed, kidney disease is one of the leading causes of mortality and morbidity in individuals with diabetes. Mortality is particularly high in patients with diabetes who also have significant proteinuria as this is associated with accelerated cardiovascular disease, further contributing to increased mortality and morbidity [1]. It is important to note that chronic kidney disease (CKD) can also be caused by other underlying conditions, such as glomerulonephritis or polycystic kidney disease [2,3,4].

The characterization and classification of DKD is based on a gradual decline in renal function, which is usually associated with increased albuminuria. DKD is often associated with elevated albuminuria [2]. The degree of albuminuria can vary between micro- or macroalbuminuria. Furthermore, diabetes is often accompanied by elevated arterial blood pressure along with increased cardiovascular complications [2].

Diabetes is the major cause for ESRD in the Western World and accounts for about 25% to 40% of all cases across the world [2].

According to the Australasian Paediatric Endocrine Group (PEG) and the National Diabetes Services Scheme (NDSS), 1.3 million (1 in 20) Australians are living with either Type 1 or Type 2 diabetes (excluding women with gestational diabetes) based on figures collated in 2020. The prevalence of diabetes linked with APEG and NDSS increased with age. Almost 19.4% (1 in 5) of Australians living with the disease are in the age group between 80 and 84, which is almost 30 times higher than those aged under 40 (0.7%). Furthermore, diabetes was more commonly found in males than in females [3].

In the Aboriginal and Torres Strait Islander communities, the prevalence of type 2 diabetes remains concerningly high. According to the 2018–19 National Aboriginal and Torres Strait Islander Health Survey (NATSIHS) conducted by the Australian Bureau of Statistics (ABS), 7.9% of Indigenous Australians were living with type 2 diabetes [3]. This figure is similar to the 7.7% reported in the previous ABS survey from 2014 [3]. The self-reported and measured data from these surveys indicate that Indigenous Australians are significantly more prone to developing diabetes compared to the general Australian population. In fact, they are approximately 3 times more likely to have diabetes [3]. Furthermore, the Aboriginal and Torres Strait Islander communities experience some of the highest rates of diabetic kidney disease (DKD) in Australia.

Chronic kidney disease (CKD), particularly diabetes-related kidney disease, is a significant global health concern, affecting approximately 1.4 million Australians and posing a risk to 6 million individuals with multiple risk factors [3]. Early diagnosis and intervention are crucial in reducing the risk of cardiovascular disease and kidney failure progression. Improving health outcomes in communities relies heavily on increasing recognition and early detection of impaired kidney function and kidney damage, including albuminuria [4]. Diabetic kidney disease (DKD) is characterized by decreased glomerular filtration rate (eGFR) and increased urinary albumin excretion, both of which contribute to elevated cardiovascular risk and mortality rates [5,6,7]. The presence and degree of urinary albumin excretion remain the cornerstone and best surrogate marker for diagnosing DKD or CKD [8], emphasizing the importance of accurate and accessible albuminuria detection methods. Albuminuria is a key biomarker for DKD and other CKD, indicating increased permeability of the glomerular basement membrane and podocyte injury [4,9,10,11]. It is categorized into three levels: normoalbuminuria (<30 mg/L), microalbuminuria (30–300 mg/L), and macroalbuminuria (>300 mg/L). The urinary albumin/creatinine ratio (UACR) is considered the gold standard for assessing albuminuria as it corrects for urinary creatinine concentration reflecting renal function [4,9,12,13,14,15].

Creatinine, a product of creatine metabolism, is used to assess renal function and muscle damage [16,17,18,19]. The estimated glomerular filtration rate (eGFR) is calculated based on serum creatinine concentration to evaluate kidney function [9,13,14,15].

Urinalysis has been a crucial diagnostic tool since ancient times [20,21,22,23]. Currently, dipstick tests are widely used for screening albuminuria due to their simplicity and speed [9,20,24,25,26,27,28,29,30,31,32]. However, they have low sensitivity (around 70%) for detecting albuminuria, with detection thresholds only reaching 10–20 mg/dL for albumin [33]. These sensitivity constraints in detecting albumin underscore a critical need in healthcare diagnostics for the development of more advanced point-of-care (POC) testing methods to address these limitations, thereby improving diagnostic accuracy and enhancing patient care outcomes. The gold standard for measuring albuminuria is the urinary albumin/creatinine ratio (ACR) [1,34,35,36], which is more reliable but requires laboratory visits and is expensive.

UV/Visible spectroscopy has been previously used to determine uric acid [37,38,39,40,41], inorganic phosphorous [42], oflaxacin [43], purple urine analysis [44], amphetamines/amines [45], creatinine crystals [46], and nitrites [47], as well as for the simultaneous determination of anti-viral drugs [48] and pregabalin [49], in urine. Stevens et al. [45] used a spectrophotometric method to screen urine for amines by facilitating a reaction between amines and carbon disulfide in warm aqueous solution under alkaline conditions. Yin et al. [42] determined inorganic phosphorous by titrating the urine samples with a buffer containing Yb^3+^ (a blue solution), simultaneously noting the changes in the UV/Visible spectrum until there were no further changes occurring denoting the end of the titration and a point where the inorganic phosphorous concentration in the urine could be calculated. Olmo et al. [43] applied UV/Visible spectroelectrochemistry to determine the ofloxacin in urine without any prior treatment of samples. Rezai et al. [39] used a method based on gold nanoparticles couples with UV/Visible spectroscopy to determine the concentration of uric acid in urine samples. Hardin et al. [44] analyzed purple urine using a NanoDrop One UV/Visible spectrometer with a 1 cm path-length cuvette. The patient’s purple urine was diluted with deionized water, and its absorption spectrum was obtained. Subsequently, methylene blue and hydroxocobalamin solutions were mixed, and the absorption spectrum of this mixture was also obtained and compared with the diluted urine sample. The mixture showed maximum absorbance peaks, while the purple urine exhibited minimum absorbance peaks.

Werle et al. [46] utilized UV/Visible spectroscopy and chromatographic analysis to examine creatinine in urine samples. They compared the absorption spectra of urine samples dissolved in distilled water with those of creatine and creatinine to detect any changes in the spectral profile. All the mentioned studies involved sample processing or pre-processing before spectroscopic analysis.

Lin et al. [41] employed a label-free method to estimate uric acid in spot urine using a portable UV/Visible instrument. They found that uric acid could be detected non-invasively without chemical labeling. Their calibration method showed R^2^ values greater than 0.95, indicating that their approach could effectively quantify uric acid in urine.

Given the limitations of current methods, there is a need for cost-effective, non-invasive techniques to measure urinary albumin excretion [37,38]. Spectroscopic approaches, such as UV/Visible spectroscopy, offer promising alternatives for point-of-care testing. This study aims to determine the Limit of Detection for a UV/Visible spectrometer approach using aqueous urine samples spiked with BSA and creatinine and to correlate the results with ELISA test values. Our spectroscopic alternative for point-of-care (POC) testing offers numerous advantages: it is high-throughput, is fast, provides real-time results, is cost-effective, and demonstrates sensitivity comparable to the gold standard ACR measurement. Additional benefits include remote testing capabilities, a reduced need for travel to urban centers, immediate results in local healthcare settings, and less invasive sampling methods. It supports early detection of conditions, effective monitoring of treatment outcomes, and minimizes loss to follow-up. Furthermore, it enables repeated testing, facilitating better risk stratification and advancing personalized medicine.

## 2. Materials and Methods

Mid-stream spot urine was collected from one healthy volunteer with ethical approval given by Monash university Ethics (2019-21431-34708). All the samples were kept at room temperature until their temperature was stable. The standard albumin (bovine serum albumin: BSA), herewith referred to as BSA (Mol Wt 66kDa), and Creatinine (CR) 113.12 g/mol were purchased from Merck (St. Louis, MO, USA). Clinically relevant concentration ranges were chosen for the BSA and CR. From one volunteer, we collected four urine samples on different days. From each sample, we prepared 13 fractions to construct spiking models for urine spiked individually with bovine serum albumin (BSA) or creatinine and 15 fractions for the co-spike (BSA/creatinine) model. Triplicate spectra were recorded for each fraction.

Bovine serum albumin (BSA) is widely used as a surrogate for human serum albumin (HSA) in experimental protocols due to its practicality and scientific validity. The two proteins share significant structural homology, with 76% sequence identity, comparable tertiary structures, similar disulfide bridge arrangements, and analogous ligand-binding sites. These similarities result in nearly identical spectroscopic properties, including UV/Visible absorption, fluorescence, and circular dichroism spectra, making BSA an effective calibration standard for albumin quantification. Additionally, BSA offers greater commercial availability, and cost-efficiency, and avoids the ethical concerns associated with human-derived materials. These attributes collectively establish BSA as an ideal substitute for HSA in spectroscopic methodologies for albumin detection.

This experiment was carried out in three stages: Y1—urine spiked with BSA, Y2—urine spiked with creatinine, and Y3—urine co-spiked with BSA and creatinine to enable the investigation of potential interactive effects between these two constituents. Each of the stages of experiments involved three biological replicates each. Each biological replicate had 12 and 15 concentration series of spiked BSA, creatinine, and BSA/creatinine, respectively, for the three stages, respectively.

### 2.1. Experiment Y1: Urine Spiked with BSA

A stock solution of 500 ppm of BSA spiked in urine was made by weighing 0.1 g of BSA in 200 mL of urine; from this stock solution, 10 mL, 9 mL, 8 mL, 7 mL, 6 mL, 5 mL, 4 mL, 3 mL, 2 mL, 1 mL, 0.5 mL, and 0.2 mL were pipetted out into 10 mL volumetric flasks labeled 1 through to 12. The pipetted stock solution in each of the 10 mL volumetric flask was diluted with 0 ppm urine up to the 10 mL mark of each volumetric flask resulting in 500 ppm, 450 ppm, 400 mg/L, 350 mg/L, 300 mg/L, 250 mg/L, 200 mg/L, 150 mg/L, 100 mg/L, 50 mg/L, 25 mg/L, and 10 mg/L concentrations, respectively, as indicated in Table 1; the volumetric flask labeled 13 contained 0 mg/L of BSA, i.e., the control urine.

### 2.2. Experiment Y2: Urine Spiked with Creatinine

A stock solution of 2000 mg/L of BSA spiked in urine was made by weighing 0.4 g of creatinine in 200 mL of urine; from this stock solution, 10 mL, 9 mL, 8 mL, 7 mL, 6 mL, 5 mL, 4 mL, 3 mL, 2 mL, 1 mL, 0.5 mL, and 0.2 mL were pipetted out into 10 mL volumetric flasks labeled 1 through to 12. The pipetted stock solution in each of the 10 mL volumetric flasks was diluted with 0 ppm urine up to the 10 mL mark of each volumetric flask resulting in 2000 mg/L, 1800 mg/L, 1600 mg/L, 1400 mg/L, 1200 mg/L, 1000 mg/L, 800 mg/L, 600 mg/L, 400 mg/L, 200 mg/L, 100 mg/L, and 40 mg/L concentrations, respectively, as indicated in Table 2; the volumetric flask labeled 13 contained the 0 ppm of creatinine, i.e., the control urine.

### 2.3. Experiment Y3: Urine Co-Spiked with BSA and Creatinine

This involved mixing of appropriate amounts of BSA and creatinine with 200 mL of urine resulting in a stock solution of co-spiked concentration of different ratios (BSA–CR). There were stock solution sets of six ratios prepared. The different ratios of stock solutions prepared were 1:2, 1:3, 1:3.75, 1:4, 1:5.2,1:6.1; from this stock solution, desired amounts were pipetted into a range of 10 mL volumetric flasks labeled 1 through to 15. The pipetted stock solution in each of the 10 mL volumetric flasks was diluted with 0 ppm urine up to the 10 mL mark of each volumetric flask resulting in a co-spiked concentration of (BSA–creatinine): 500:2000 mg/L, 450:1800 mg/L, 400:1600 mg/L, 350:1050 mg/L, 300:900 mg/L, 230:1400 mg/L, 250:1300 mg/L, 211:1100 mg/L, 150:780 mg/L, 180:625 mg/L, 138:481 mg/L, 90:312 mg/L, 50:100 mg/L, 100:200 mg/L, and 25:75 mg/L concentrations, as indicated in Table 3; the volumetric flask labeled 16 contained the 0 ppm of BSA and creatinine, i.e., the control urine.

### 2.4. Instrument Parameters and Experimental Procedure

The NanoDrop One UV/Visible spectrometer from Thermo Fisher Scientific requires just 1–2 µL of sample for accurate protein quantification. Its innovative sample retention technology eliminates the need for dilutions, even with highly concentrated samples. This allows for precise measurement of sample concentration without prior knowledge as it always falls within the instrument’s dynamic range.

To measure, the sample is placed on the pedestal, and the upper sampling arm is lowered to initiate the spectra measurement. The surface tension properties of the pedestal hold the sample in place between the two optical fibers. A xenon lamp sends light through the top optical fiber, which passes down through the sample column and is detected by the internal spectrometer [50].

Path length is a crucial factor in Beer’s Law and is considered when calculating sample concentration. In this instrument, the path length is the distance between the optical fibers on the upper and lower surfaces. Although the lower detection limit of the internal spectrometer is approximately 1.5 absorbance units, the sample retention technology allows for shorter path lengths, extending the absorbance measurement range. The instrument features a dynamic path length range from 0.03 to 1.0 mm, auto-ranging.

For example, using path lengths shorter than the standard 10 mm cuvette allows for the measurement of more concentrated samples. A 10 mm path length has a maximum concentration of up to 75 ng/µL dsDNA (with a maximum absorbance of 1.5), while a 0.2 mm path length can measure concentrations up to 3750 ng/µL, normalized to a 10 mm path length corresponding to a maximum absorbance value of 75.

All fractions were measured in triplicate to ensure reproducibility. Prior to each sample measurement, a blank spectrum was recorded using 1 µL of water as a background reference. This procedure ensured accurate baseline correction and minimized potential interference from environmental factors.

### 2.5. Data Analysis and Pre-Processing

Chemometric analysis was employed to determine the concentrations of BSA and creatinine from the spectral data following appropriate spectral pre-processing. Regression models were developed using MATLAB R2024b (MathWorks, Natick, MA, USA) in combination with the PLS Toolbox (Eigenvector Research, Manson, WA, USA). The PLS Toolbox, a specialized MATLAB add-on, provides an extensive suite of tools for advanced chemometric analysis, including Partial Least Squares (PLS) regression modeling, spectral processing, data pre-processing, exploratory analysis, and model validation and optimization.

The data were divided into calibration and validation sets to ensure robust model development and evaluation. Two urine samples from different days were used for calibration, while one urine sample collected on another day was reserved for validation. For the urine samples spiked individually with BSA and creatinine, the combined calibration and validation sets consisted of 39 fractions. For the co-spiked (BSA/creatinine) experiment, the calibration and validation sets included a total of 45 fractions. Specifically, for urine spiked individually with BSA or creatinine, the calibration set contained 26 fractions, and the validation set contained 13 fractions. For the co-spiked experiment, the calibration set included 30 fractions, while the validation set comprised 15 fractions. Each fraction was measured in triplicate.

The predictive performance of the calibration models was assessed using the independent validation set. The calibration and validation datasets were organized into matrices, denoted as Z and Z_test, respectively. The matrices were structured with dimensions A × B for the calibration set and C × B for the validation set, where A and C represent the number of samples, and B corresponds to the number of absorbance wavelengths. The concentrations of BSA or creatinine were subsequently predicted from these datasets, allowing the evaluation of model accuracy and reliability.

Pre-processing steps included the following:Baseline correction (weighted least squares automatic baseline removal) to eliminate baseline effects and enable raw spectra comparison.Smoothing to reduce noise.Standard Normal Variate (SNV) normalization to account for path length differences due to varying refractive properties of urine samples.Mean centering to highlight spectral differences by subtracting the mean spectrum from each spectrum.

After pre-processing, Partial Least Squares Regression (PLS-R) was applied. PLS-R utilizes the relationship between predictors (spectra) and predicted values (concentrations), capturing maximum variables and providing covariance information. PLS-R comprises two matrices: loadings and scores. It is an iterative process that finds vectors best explaining the variance, which are then plotted against each other to provide a linear model. The R^2^ value represents the square of the correlation of the calibration data.

We conducted a 4-fold cross-validation using a Venetian blinds approach, ensuring that no technical replicates were included in the calibration set. As a result, the samples used for testing were not part of the calibration set and were effectively independent of the modeling process.

For PLS-R calibration plot quantification, spectral trimming was applied, focusing on the range from 190 nm to 320 nm.

## 3. Results and Discussion

Figure 1 presents the UV/Visible absorption spectra of bovine serum albumin (BSA) and creatinine in water at a concentration of 150 mg/L and a spectrum of unspiked urine. The spectrum for BSA features a prominent peak at 217 nm, which corresponds to the peptide (amide) chromophore of the protein, primarily associated with the side chains of tryptophan (W) and phenylalanine (F). Additionally, a peak at 280 nm arises from the combined contributions of the main light-absorbing amino acids: tryptophan (W), tyrosine (Y), and phenylalanine (F). Although other light-absorbing amino acids are present in proteins, their absorption intensities in this spectral range are less pronounced compared to W, Y, and F [41,51,52,53,54,55]. These other amino acids generally exhibit more significant absorption bands in the far and middle ultraviolet regions, which are typically not analyzed in this context.

Creatinine, on the other hand, displays distinct peaks at 219 nm, 229 nm, and 248 nm. The urine spectrum reveals characteristic bands at 230 nm and 248 nm, associated with naturally occurring creatinine found in urine samples. This spectral analysis establishes a basis for differentiating and quantifying BSA and creatinine in urine, which is essential for evaluating albuminuria and kidney function

### 3.1. Y1—Urine Spiked with BSA

The first experiment involved adding bovine serum albumin (BSA) to the urine sample at 13 different concentrations. The resulting spectra, Partial Least Squares Regression (PLSR) plot, and regression vector plot are illustrated in Figure 2a, Figure 2b, and Figure 2c, respectively.

UV/Visible spectroscopy is particularly sensitive to chromophoric molecules. Among the 20 amino acids, only three are aromatic: Tryptophan, Tyrosine, and Phenylalanine. These amino acids exhibit prominent absorbance bands due to π to π* transitions originating from their aromatic ring structures (the indole group of tryptophan, the phenolic side chain of tyrosine, and the mono-substituted benzene structure of phenylalanine), as well as their peptide bonds.

In Figure 2a, absorption bands are observed at 230 nm, 243 nm, 248 nm, 257 nm, and 287 nm, with an isosbestic point at 254 nm. The peak at 230 nm is attributed to the tryptophan residue of albumin, while the peaks at 243 nm, 248 nm, 254 nm, and 257 nm correspond to tyrosine and phenylalanine. The peak at 287 nm reflects a combined contribution from tyrosine and tryptophan residues. The intensity of these bands depends on their extinction coefficients (ε). Although tyrosine and phenylalanine also absorb at 230 nm, their contributions are less significant compared to tryptophan, which has an extinction coefficient of 47 M^−1^ cm^−1^.

From 190 nm to 254 nm (the isosbestic point for phenylalanine), absorbance increases as BSA concentration in spiked urine fractions rose from 0 to 500 ppm. However, beyond the isosbestic point, a different trend is observed: absorbance decreases as BSA concentration increases.

The isosbestic point at 254 nm, associated with phenylalanine, indicates a transition between two distinct states or conformations of the protein. At this point, the phenylalanine residue may be in a buried or exposed state. In the buried state, the aromatic ring structure of phenylalanine engages in hydrophobic interactions with the protein and other peptide side chains, effectively shielding itself from the solvent. Conversely, in the exposed state, the aromatic benzene ring can interact with solvent molecules through hydrogen bonding, dipole interactions, ion–dipole interactions, or van der Waals forces. Therefore, the protein spectra are influenced by the surrounding environment, with the aromatic ring structure of phenylalanine playing a crucial role in determining the absorption bands observed in the spectra.

After pre-processing, the data underwent Partial Least Squares Regression (PLS-R). This method analyzes the relationship between the predictor variables (spectra) and the predicted variable (concentration), maximizing the capture of variance and providing covariance information. PLS-R consists of two matrices: loadings and scores, and it employs an iterative process to identify vectors that best explain the data’s variance.

#### 3.1.1. Calibration Plot Analysis

Figure 2b presents the PLS-R calibration plot, which displays two vectors plotted against each other, facilitating the construction of a linear model. The R^2^ value, which indicates the square of the correlation for the calibration data, reflects the model’s accuracy. The PLS-R calibration plot, based on five latent variables and generated using MATLAB, achieved R^2^ values of 0.92 for calibration, 0.87 for cross-validation, and 0.92 for prediction (PRED(CV)). These values demonstrate a strong correlation between the spectra and BSA concentration, highlighting the model’s effectiveness in explaining data variability. The Limit of Detection (LOD) for BSA using this method was calculated to be 28.23 mg/L.

#### 3.1.2. Regression Vector Plot Insights

The regression vector plot in Figure 2c illustrates the absorption bands responsible for the calibration. A subtle positive band at 243 nm correlates with higher BSA concentrations, while negative bands at 221 nm, 229 nm, and 257 nm are associated with lower BSA concentrations in urine spiked fractions. The band at 254 nm, which lies between the positive and negative axes, contributes to the calibration plot. Its position at the isosbestic point, where a conformational transition of the protein occurs, underscores its significance in the regression vector.

The labeled bands in the regression vector plot align well with the bands of BSA in water; however, a slight red shift is observed in the band positions within urine. This shift can be attributed to various factors, including the protein–solvent environment. It highlights the importance of considering matrix effects when developing spectroscopic methods for complex biological samples like urine.

### 3.2. Y2—Urine Spiked with Creatinine

The second experiment focused on adding creatinine to urine at 13 different concentrations. The resulting spectra, Partial Least Squares Regression (PLSR) plot, and regression vector plot are illustrated in Figure 3a, Figure 3b, and Figure 3c, respectively.


**Spectral Analysis**


Prominent absorption bands at 229 nm and 249 nm correspond to creatinine and align well with the bands observed in creatinine dissolved in water (see Figure 1). In Figure 3a, the 220–250 nm region shows a hypochromic shift for the bands at 249 nm and 229 nm as the creatinine concentration increases.

Two isosbestic points for creatinine in urine are noted at 294 nm and 296 nm. At the 294 nm isosbestic point, a shift in peak intensities occurs—demonstrating both hyperchromic and hypochromic shifts—between lower and higher concentrations. This pattern reverses at 296 nm, functioning like a toggle switch. These phenomena may be attributed to the interactions between creatinine and other ions or proteins in the urine, highlighting the complexity of the solvent environment. Additionally, they may indicate slight changes in urine pH due to increasing creatinine concentration.

Creatinine, being a guanidine-like compound, typically exhibits peaks at 281 nm and 287 nm in simple solutions, which are pH-dependent. As concentration increases, a peak in the >290 nm region emerges. However, in the urine spectra, these bands shift to longer wavelengths by 2–5 nm compared to non-urine solutions, indicating the significant influence of urine’s complex matrix on spectral characteristics.

#### 3.2.1. Insights and PLSR Results

The observed spectral shifts and isosbestic points provide valuable insights into creatinine’s behavior in urine, which is essential for developing accurate quantification methods. These spectral features can be leveraged in multivariate analysis techniques, such as PLSR, to create robust calibration models for determining creatinine in urine samples.

Figure 3b presents the PLSR plot for urine spiked with creatinine, explained by four latent variables. The model demonstrates strong performance, with R^2^ values of 0.97, 0.96, and 0.98 for calibration, cross-validation, and prediction, respectively. These high R^2^ values indicate a strong correlation between the spectral data and creatinine concentrations, underscoring the model’s effectiveness in quantifying creatinine in urine samples.

#### 3.2.2. Regression Vector Plot Analysis

Figure 3c displays the regression vector plot, illustrating the impact of individual predictor variables on the response variable. Negative peaks in this plot correspond to bands associated with lower creatinine concentrations. The peak at 249 nm in the regression vector aligns closely with the 248 nm band observed for creatinine in water. The 1 nm difference can be attributed to interactions between creatinine and other components in the complex urine matrix.

Additionally, a broad, fused band at 294 nm and 296 nm is evident in the regression vector plot, indicating the isosbestic points of creatinine in urine. These points likely reflect conformational changes of creatinine between its free and bound forms.

The isosbestic point of creatinine is expected to depend on the ionic strength of the urine. Creatinine is likely present not as a neutral molecule but in various ionic forms due to the presence of certain carboxylic groups in urine. The cationic form of creatinine may play a crucial role in binding H^+^ ions in urine. Furthermore, the carboxylic group could exhibit bipolar characteristics, resulting in an overall isoelectric molecule. Alternatively, it could represent the dissociation of the guanidine molecule to a neutral point, leading to differing bound conformations and associations.

The spectral characteristics and their interpretations provide critical insights into creatinine behavior in urine, essential for developing accurate and reliable quantification methods. The high performance of the PLSR model, combined with detailed spectral analysis, highlights the potential of this approach for precise creatinine determination in complex urine samples.

### 3.3. Y3-Urine Co-Spiked with Bovine Serum Albumin (BSA) and Creatinine (CR)

In the third experiment, spectral trimming was utilized to process the data from urine spectra co-spiked with BSA and CR. Two prominent bands at 229 nm and 249 nm correspond to BSA and CR, respectively, as discussed in the previous sections on Ur/BSA and Ur/CR. Although most other bands are convoluted, they can be distinctly identified in the individual regression vector plots.

#### 3.3.1. Performance of PLS-R Models

The Partial Least Squares Regression (PLS-R) plot for BSA (Figure 4b) shows Root Mean Square Error of Calibration (RMSEC) and Root Mean Square Error of Cross-Validation (RMSECV) values of 66.93 mg/L and 73.92 mg/L, respectively. For CR (Figure 5b), the RMSEC and RMSECV values are 244.32 mg/L and 275.65 mg/L, respectively. The prediction models for both BSA and CR demonstrate robust performance, with R^2^ PRED values of 0.96 and 0.95, indicating strong reliability as these values approach unity.

#### 3.3.2. Regression Vector Analysis

In the BSA regression vector plot (Figure 4c), the bands at 221 nm, 229 nm, and 249 nm (attributable to existing CR in urine) contribute to the calibration plot (Figure 4b). The 221 nm and 229 nm bands corroborate with those observed in the urine spiked with BSA model (Figure 2c), confirming their origin from BSA.

For CR, the regression vector plot (Figure 5c) displays bands at 249 nm, 287 nm, and 296 nm. The distinct 249 nm band aligns well with the corresponding band in Figure 3c (urine spiked with CR). The 287 nm and 296 nm bands correspond to CR bands seen in the spectra (Figure 3a). Notably, the 296 nm band is absent in the spectra and regression vector plot of urine spiked with BSA (Figure 2a,c), highlighting the successful chemometric decomposition of the dataset and the separate elucidation of both BSA and CR analytes in the co-spiked experiment.

#### 3.3.3. Key Observations

A significant observation is the presence of the 249 nm band in both Figure 4c and Figure 5c, which originates from CR in both cases. The primary difference is in band intensity, which is markedly higher in Figure 5c due to the greater CR concentration in that model. This 249 nm band correlates with the band observed in Figure 3c (urine spiked with CR), while its presence in Figure 2c (urine spiked with BSA) is attributed to the background concentration of CR in unspiked urine, albeit at a lower intensity compared to samples spiked or co-spiked with CR.

These observations, particularly the differences in band intensities in the regression vector plots, reaffirm the accurate extraction of BSA and CR from the co-spiked model. This precise decomposition is ultimately reflected in the construction of the PLS-R plot, demonstrating the effectiveness of the spectral analysis and chemometric approach in quantifying multiple analytes in complex urine samples.

### 3.4. Limit of Detection (LoD) and Limit of Blank (LoB)

The Limit of Detection (LoD) and Limit of Blank (LoB) were calculated from the predicted dataset after data pre-processing. These parameters are crucial for assessing the analytical performance of the method. The LoB is an important parameter defined as the highest apparent analyte concentration expected when analyzing a blank sample. It is determined using the following formula [56]:LoB = mean(blank) + 1.645 × SD(blank)
where mean(blank) is the average of the blank dataset measurements, and SD(blank) is the standard deviation of these measurements.

The LoD is used to determine the lowest concentration of the analyte that can be reliably detected using the employed methods. It is calculated by incorporating information from both the blank samples and samples containing the lowest concentration of the analyte:LoD = LoB + 1.645 × SD(low concentration sample)

Table 4 presents the experiment’s measurement of absolute sensitivity, represented by the LoD values. Due to the different clinically relevant ranges for BSA and creatinine (CR), we observe a higher LoD for creatinine compared to BSA. The maximum concentrations used in this study were 2000 mg/L for creatinine and 500 mg/L for BSA, while the minimum concentrations were 40 mg/L for creatinine and 10 mg/L for BSA.

It is important to note that the lowest concentration corresponding to the predicted values derived from the model is used for calculation in the LoD formula. This approach ensures that the LoD reflects the method’s capability to detect the analyte at concentrations near the lower end of the calibration range.

The difference in LoD values between BSA and CR can be attributed to several factors:The distinct spectral characteristics of each analyte;The varying degrees of spectral overlap with other urine components;The different concentration ranges used in the calibration models.

These LoB and LoD calculations provide valuable information about the method’s sensitivity and its ability to distinguish between the presence and absence of the analytes in urine samples. They are essential metrics for validating the analytical performance of the developed spectroscopic method for simultaneous quantification of BSA and creatinine in urine.

The significance of the LOD’s achieved as indicated table below are much below the normal physiological range of 0–30 mg/L for albumin and 600–1800 mg/L for creatinine.

The gold standard to measure albuminuria or proteinuria is the urinary albumin/creatinine ratio (ACR). This ratio corrects the urinary albumin concentration for the urinary creatinine ratio, which is dependent on the urinary excretion of creatinine. This is directly determined by renal function and also corrects for different hydration levels. At low creatinine concentrations (near or below the LOD of 58.43 mg/L), the denominator in the ACR calculation becomes less precise due to higher uncertainty or noise in creatinine measurements. This can lead to significant errors or variability in the calculated ACR, particularly for samples near the lower physiological range of creatinine (600–1800 mg/L). The creatinine LOD of 58.43 mg/L is sufficient for most normal physiological ranges but may introduce challenges in accurately calculating the ACR for samples with low creatinine concentrations. This could result in reduced precision or misclassification in certain clinical scenarios, particularly for patients with low urinary creatinine output (creatinine is high in the serum) reflecting reduced renal function. Reducing the LOD for creatinine or enhancing the method’s sensitivity would help improve the robustness of ACR measurements at lower concentrations.

Our experiments utilized human spiked urine samples, which we consider advantageous over artificially spiked urine, as real human urine more closely resembles clinical samples. Unlike artificial urine, real human urine contains natural metabolites and biological components similar to those found in clinical specimens. However, clinical samples may also include medications that can introduce drug-associated spectral effects, alter pH levels, and cause conformational changes in albumin, all of which could potentially impact the results.

We acknowledge the limitations of using spiked samples. These samples involve controlled concentrations of purified standards and analytes, which do not fully reflect the natural variability and metabolite ratios present in clinical specimens. Consequently, spiked samples may not adequately assess the robustness of the method against biological variability and can underestimate the effects of matrix interferences. To address these limitations, future studies should incorporate clinical specimens to evaluate the method’s performance under real-world conditions, accounting for the complexities introduced by medications, pH variations, and other biological interferences.

Given the current high cost of laboratory-based albumin/creatinine ratio (ACR) testing using ELISA (USD $20 per test), we propose a miniaturized diagnostic device to complement existing testing platforms. This device is designed to enhance accessibility and improve testing efficiency across diverse healthcare settings. While miniaturized devices may not match the analytical precision of conventional large laboratory equipment, their ability to expand diagnostic reach represents a valuable trade-off, particularly in resource-limited contexts.

The spectral data generated by the device can be connected to a cloud-based platform via a smartphone, enabling medical record integration and eHealth applications. Using machine-based algorithms, the data can be securely analyzed and transmitted to healthcare providers. The diagnosis is then communicated to both the patient and healthcare professionals, allowing for timely clinical decision-making.

To ensure the method’s reliability, validation will involve rigorous testing protocols, including parallel testing with the validated ELISA method through split-sample analysis, blinded testing, inter-laboratory comparisons, and cross-validation with reference laboratories. This comprehensive approach will help establish the accuracy, reproducibility, and robustness of the proposed diagnostic platform.

## 4. Conclusions

In this study, we employed UV/Visible spectroscopy with the NanoDrop One to investigate the spectral features of urine samples spiked with bovine serum albumin (BSA) and creatinine, including co-spiked samples, without any prior sample processing, marking the first time that this approach has been used to assess disease markers in urine. Our findings reveal that higher concentrations of BSA and creatinine lead to elevated and shifted absorbance patterns. We achieved excellent correlation values (R^2^ and R^2^ PRED) for each model, with regression vector plots illustrating distinct spectral bands associated with BSA and creatinine, along with their corresponding amino acid bands.

This indicates the potential of this technique for detecting pathological conditions in patients with diabetes, diabetic kidney disease (DKD), and chronic kidney disease (CKD).

This novel, label-free spectroscopic approach, which requires no sample pre-processing and is enhanced by data fusion, could serve as a real-time point-of-care (POC) diagnostic tool for measuring urinary albumin in the future. Using single wavelength diodes, the technology could be integrated into a compact, enabling secure data transfer to smartphones and cloud platforms for eHealth and medical record integration. This facilitates efficient communication with healthcare providers and supports machine learning applications, allowing for prompt delivery of diagnoses to patients and healthcare professionals.

Such a device would offer significant advantages: it is cost-effective, fast, and robust and provides immediate results with remote, decentralized access. The device can be reused, facilitating safe monitoring of treatment effects in an economical manner. This capability helps identify risk levels and supports individualized medical care.

By transforming traditional lab-based spectroscopic equipment into a practical point-of-care instrument, our approach complements routine blood tests used in specialized diabetes centers both nationally and internationally. It is particularly well suited for deployment in remote areas of Australia, including Indigenous populations at high risk for CKD. Furthermore, it would enable safe POC screening of pregnant women with diabetes for early kidney disease. Immediate access to eHealth and medical records via the cloud will enhance healthcare delivery by facilitating earlier and more targeted treatments.

## Figures and Tables

**Figure 1 sensors-25-03307-f001:**
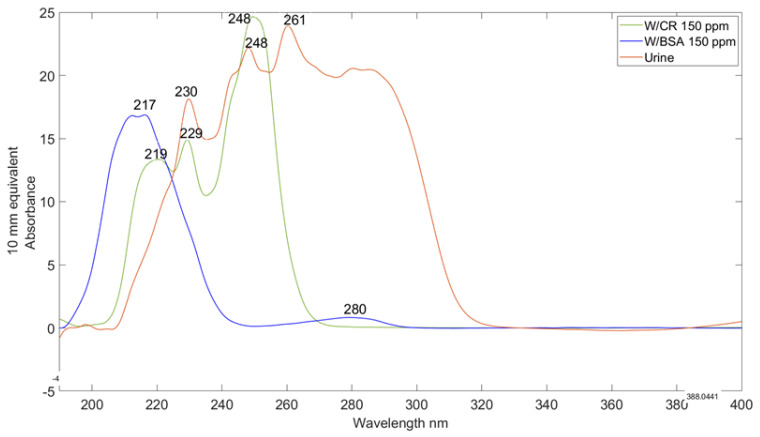
Spectra of CR, BSA spiked in water, and unspiked urine recorded using the NanoDrop UV/Visible spectrometer. The concentration of BSA and CR in water was 150 mg/L.

**Figure 2 sensors-25-03307-f002:**
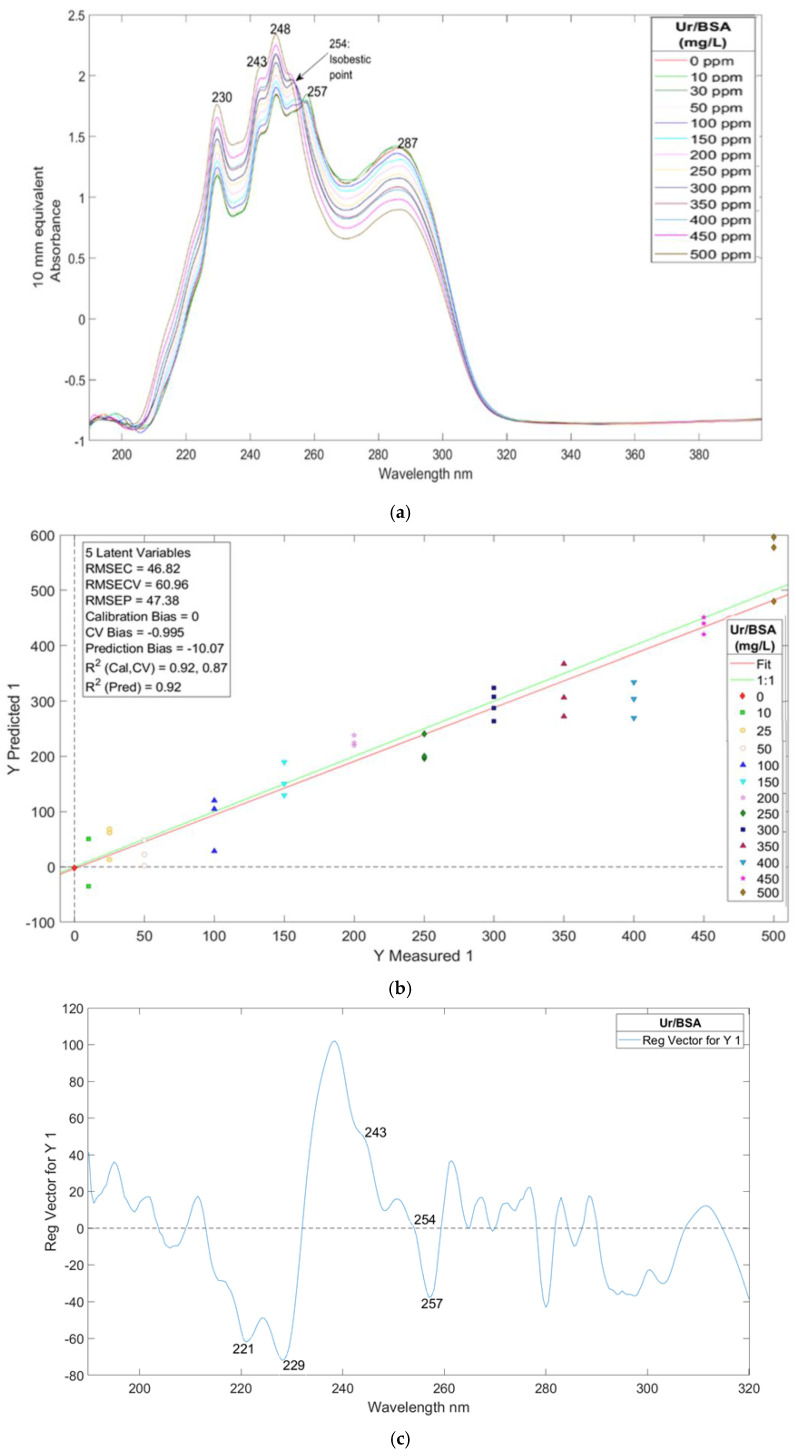
(**a**) UV/Visible spectra of spiked urine with different amounts of BSA of sample Y1. (**b**) PLS-R calibration plot prediction model of urine spiked with BSA of sample Y1. (**c**) Regression vector of latent variable 4 from the prediction model of sample Y3 of urine spiked with BSA.

**Figure 3 sensors-25-03307-f003:**
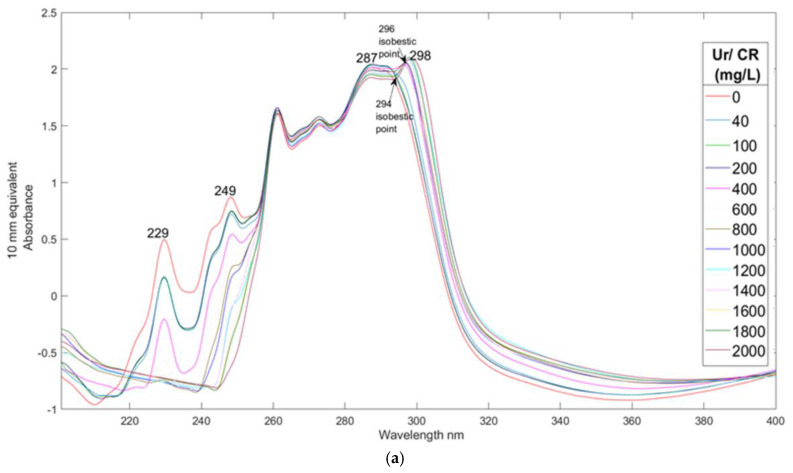
(**a**) UV/Visible spectra of spiked urine with different amounts of creatinine of sample Y2. (**b**) PLS-R calibration plot prediction model of urine spiked with creatinine of sample Y2. (**c**) Regression vector of latent variable 4 from the prediction model of sample Y2 urine spiked with CR.

**Figure 4 sensors-25-03307-f004:**
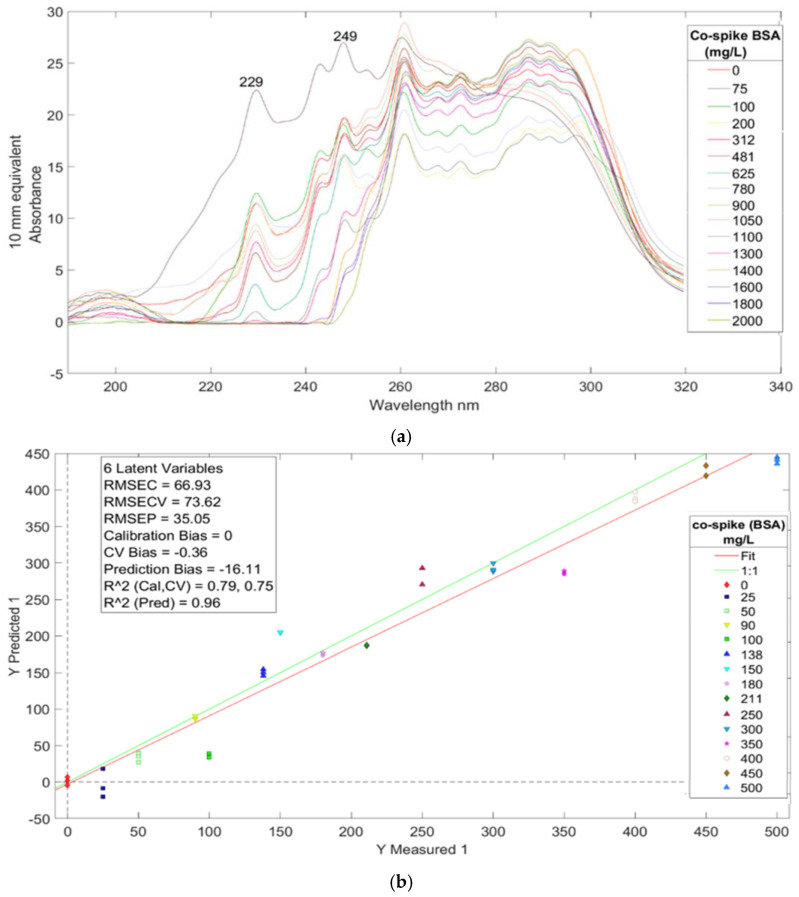
(**a**) UV/Visible spectra of co-spiked BSA/CR (BSA) urine of sample Y3. (**b**) PLS-R calibration plot prediction model for BSA of co-spiked BSA/CR urine sample Y3. (**c**) Regression vector of latent variable 6 from the prediction model of sample Y3 co-spiked BSA/CR (BSA).

**Figure 5 sensors-25-03307-f005:**
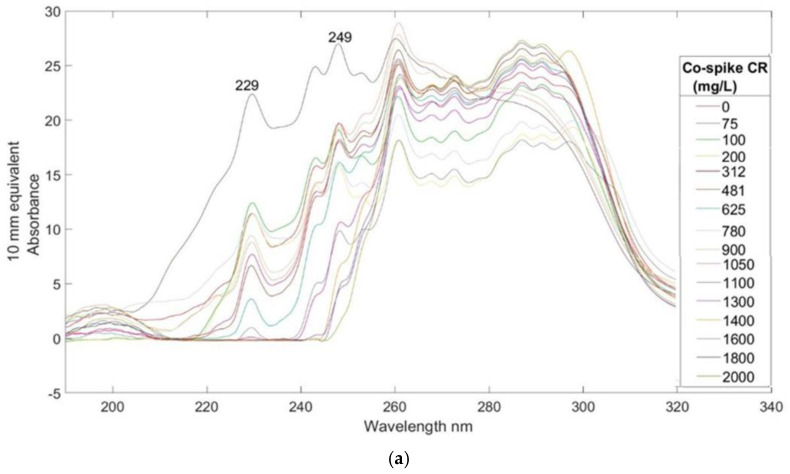
(**a**) UV/Visible spectra of co-spiked BSA/CR (CR) urine of sample Y3. (**b**) PLS-R calibration plot prediction model for CR of co-spiked BSA/CR urine sample Y3. (**c**) Regression vector of latent variable 4 from the prediction model of sample Y3 co-spiked BSA/CR (CR).

**Table 1 sensors-25-03307-t001:** Concentration table of urine spiked with BSA in mg/L.

Sample	BSA Concentration (mg/L)
1	500
2	450
3	400
4	350
5	300
6	250
7	200
8	150
9	100
10	50
11	25
12	10
13	0

**Table 2 sensors-25-03307-t002:** Concentration table of urine spiked with creatinine in mg/L.

Sample	Creatinine (CR) Concentration (mg/L)
1	2000
2	1800
3	1600
4	1400
5	1200
6	1000
7	800
8	600
9	400
10	200
11	100
12	40
13	0

**Table 3 sensors-25-03307-t003:** Concentration table of co-spiked urine with BSA and creatinine in mg/L.

Sample	BSA Concentration (mg/L)	Creatinine Concentration (mg/L)
1	500	2000
2	450	1800
3	400	1600
4	350	1050
5	300	900
6	230	1400
7	250	1300
8	211	1100
9	150	780
10	180	625
11	138	481
12	90	312
13	50	100
14	100	200
15	25	75

**Table 4 sensors-25-03307-t004:** Limit of Detection (LoD) of urine sample.

Samples	LOD mg/L
Ur/BSA	28.23
Ur/CR	39
Co-spike (BSA)	19.82
Co-spike (CR)	58.43

## Data Availability

Data for this article, including Tables of pre-processed UV/Vis spectroscopic data for experiments are available at Zenodo at https://doi.org/10.5281/zenodo.14853849.

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
