# Peer review of "Rapid, Precise, and Clinically Relevant Quantification of Urinary Albumin and Creatinine Using a NanoDrop UV/Vis Spectrophotometer"

_sensors, 2025, doi:10.3390/s25113307_

Round 1
Reviewer 1 Report
Comments and Suggestions for Authors
The study investigates a quite interesting topic. I believe the topic discussed in this paper would be of high interest to clinicians scientist and practitioners. However, the paper in its current form needs significant improvement.
#Abstract:
(1) Can the authors clearly state the clinical significance of the achieved LODs (e.g., BSA LOD of 19.82 mg/L is below the microalbuminuria threshold of 30 mg/L).
(2) Moreover, would you mind clarifying how the creatinine LOD (58.43 mg/L) impacts ACR accuracy, especially at low concentrations?
#Introduction:
(1) Please can you strengthen your rationale by explicitly linking the limitations of current methods (e.g., dipstick sensitivity, lab-based ACR costs) to the proposed UV-Vis approach?
(2) Can the authors clearly emphasize how this method addresses gaps in POC diagnostics for underserved populations like Indigenous Australians.
#Methods
(1) Pleas clarify the cross-validation protocol (e.g., k-fold, leave-one-out) used in PLS-R. Moreover, specify whether biological replicates were from different donors or aliquots from the same sample.
(2) Can the authors justify the choice and use of BSA instead of human serum albumin (HSA), as BSA may differ spectroscopically. Please address potential matrix effects in real urine vs. spiked samples.
#Results
Can the authors significantly discuss the clinical relevance of the creatinine LOD (58.43 mg/L) in co-spiked samples? If ACR is calculated as albumin (mg/L)/creatinine (g/L), explain whether this LOD is sufficient for accurate ratio determination in early-stage DKD?
#Discussion:
(1) Why didn't the authors Acknowledge the limitations, such as the use of spiked samples rather than clinical specimens? Can you address how interferences (e.g., pH, metabolites, medications) in real urine might affect results?
(2) Can the authors expand on the pilot trial design (mentioned in the abstract) to outline how clinical validation will be conducted, including comparison with ELISA and demographic diversity?
Comments on the Quality of English LanguageThe quality of English language is good, however small mistakes shall be revised. Please ensure consistent terminology (e.g., "UV-Vis" vs. "UV/Visible").
Author Response
Reviewer 1:
#Abstract:
(1) Can the authors clearly state the clinical significance of the
achieved LODs (e.g., BSA LOD of 19.82 mg/L is below the
microalbuminuria threshold of 30 mg/L).
The significance of achieved LOD is that it is much below the normal physiological range of 0-30 mg/L for albumin and 600-1800 mg/L for creatinine
The following changes have been made to the manuscript:
Abstract: Albuminuria is a sensitive biomarker of kidney dysfunction, and the Albu-min/Creatinine Ratio (ACR) is an essential measure for monitoring diabetic kidney disease (DKD). Abnormal levels can indicate a propensity for progressive renal failure and other complications such as cardiovascular diseases. This study employed UV/Visible spectros-copy to analyse aqueous urine samples spiked with bovine serum albumin (BSA) and creatinine at clinically relevant concentrations (0-30 mg/L for albumin and 600-1800 mg/L for creatinine) using multivariate method. UV/Visible spectra of co-spiked samples rec-orded in triplicate revealed distinct bands at 229 nm and 249 nm, corresponding to BSA and creatinine, respectively, alongside other amino acid bands. Partial Least Squares Regression (PLS-R) analysis for BSA yielded Root Mean Square Error of Calibration (RMSEC) and Cross-Validation (RMSECV) values of 66.93 and 73.92 mg/L, respectively. For creatinine, RMSEC and RMSECV values were 244.32 and 275.65 mg/L, respectively. Prediction models for both BSA and creatinine compared to ELISA demonstrated robust performance with R²PRED values of 0.96 and 0.95, respectively, indicating strong model reliability. The Limit of Detection (LOD) for co-spiked samples was 19.82 mg/L for BSA and 58.43 mg/L for creatinine. The significance of the achieved limit of detection (LOD) lies in its ability to measure concentrations well below the normal physiological ranges of 0–30 mg/L for albumin and 600–1800 mg/L for creatinine. These results demonstrate proof of concept of applying UV/Visible spectroscopy-based method as a rapid, cost-effective point-of-care (PoC) tool for ACR measurement, offering promising applications in the early diagnosis, monitoring, and prognosis of diabetic kidney disease and as-sociated cardiovascular complications. The next stage will involve a pilot trial to evaluate the technology's potential using clinical patients.
(2) Moreover, would you mind clarifying how the creatinine LOD
(58.43 mg/L) impacts ACR accuracy, especially at low
concentrations?
This is a very good point!
The gold standard to measure albuminuria or proteinuria is the urinary albumin/creatinine ratio (ACR). This ratio corrects the urinary albumin concentration for the urinary creatinine ratio, which is dependent on the urinary excretion of creatinine. This is directly determined by renal function and also corrects for different hydration levels. At low creatinine concentrations (near or below the LOD of 58.43 mg/L), the denominator in the ACR calculation becomes less precise due to higher uncertainty or noise in creatinine measurements. This can lead to significant errors or variability in the calculated ACR, particularly for samples near the lower physiological range of creatinine (600–1800 mg/L). The creatinine LOD of 58.43 mg/L is sufficient for most normal physiological ranges but may introduce challenges in accurately calculating the ACR for samples with low creatinine concentrations. This could result in reduced precision or misclassification in certain clinical scenarios, particularly for patients with low urinary creatinine output (creatinine is high in the serum) reflecting reduced renal function. Reducing the LOD for creatinine or enhancing the method's sensitivity would help improve the robustness of ACR measurements at lower concentrations.
#Introduction:
(1) Please can you strengthen your rationale by explicitly linking
the limitations of current methods (e.g., dipstick sensitivity, lab-based
ACR costs) to the proposed UV-Vis approach?
Dipsticks have been widely adopted as diagnostic tools for multiple analyte measurements, their performance limitations in sensitivity, present significant clinical challenges. This is very evident in albuminuria testing, where sensitivity rates are as low as 69.4%, with detection thresholds only reaching 10-20 mg/dL for albumin. These sensitivity constraints in detecting albumin underscores a critical need in healthcare diagnostics for the development of more advanced point-of-care (POC) to overcome these limitations leading to improved diagnostic accuracy and patient care outcomes.
(2) Can the authors clearly emphasize how this method addresses
gaps in POC diagnostics for underserved populations like
Indigenous Australians.
Our spectroscopic alternative for point-of-care (POC) testing offers several key advantages: it is high-throughput, fast, cost-effective, and provides real-time results, with sensitivity comparable to the gold standard ACR measurement. Additional benefits include remote testing capabilities, minimizing the need for travel to urban centers, and enabling immediate results in local healthcare settings. The less invasive sampling methods facilitate early detection of conditions, monitoring of treatment effects, and improved patient compliance by reducing loss to follow-up. Furthermore, the ability to perform repeated testing supports better risk stratification and paves the way for personalized medicine.
The following changes have been reflected in the manuscript regarding these two comments:
Urinalysis has been a crucial diagnostic tool since ancient times [20-23]. Currently, dipstick tests are widely used for screening albuminuria due to their simplicity and speed [9,20,24-32]. However, they have low sensitivity (around 70%) for detecting albuminuria, with detection thresholds only reaching 10-20 mg/dL for albumin [56]. These sensitivity constraints in detecting albumin underscore a critical need in healthcare diagnostics for the development of more advanced point-of-care (POC) testing methods to address these limitations, thereby improving diagnostic accuracy and enhancing patient care outcomes.
Given the limitations of current methods, there is a need for cost-effective, non-invasive techniques to measure urinary albumin excretion [36,37]. Spectroscopic approaches, such as UV/Visible spectroscopy, offer promising alternatives for point-of-care testing. This study aims to determine the limit of detection for a UV/Visible spectrometer approach using aqueous urine samples spiked with BSA and creatinine, and to correlate the results with ELISA test values. Our spectroscopic alternative for point-of-care (POC) testing offers numerous advantages: it is high-throughput, fast, provides real-time results, is cost-effective, and demonstrates sensitivity comparable to the gold standard ACR measurement. Additional benefits include remote testing capabilities, reduced need for travel to urban centres, immediate results in local healthcare settings, and less invasive sampling methods. It supports early detection of conditions, effective monitoring of treatment outcomes, and minimizes loss to follow-up. Furthermore, it enables repeated testing, facilitating better risk stratification and advancing personalized medicine.
[56] O. T. Browne and S. Bhandari, Bmj, 2012, 344, e2339.
#Methods
(1) Please clarify the cross-validation protocol (e.g., k-fold, leaveone-
out) used in PLS-R. Moreover, specify whether biological
replicates were from different donors or aliquots from the same
sample.
Thank you for noticing this as we did not specify the type of PLS-R cross validation that was performed.
The following changes have been made in the manuscript
We conducted a 4-fold cross-validation using a Venetian blinds approach, ensuring that no technical replicates were included in the calibration set. As a result, the samples used for testing were not part of the calibration set and were effectively independent of the modelling process.
(2) Can the authors justify the choice and use of BSA instead of
human serum albumin (HSA), as BSA may differ spectroscopically.
Please address potential matrix effects in real urine vs. spiked
samples.
BSA is used in instead of HSA in numerous scientific studies because of its easier accessibility, cost and the subsequent results are transferred to HSA. BSA has 583 amino acids in comparison to HSA that has 585 amino acids, BSA has two Tryptophan residues and HSA has a unique amino acid. BSA and HSA are similar homologs. There is a large number of similarities between BSA and HSA owing to the slight differences in their amino acid sequences. BSA is much more economically viable to use than HSA, which is hundreds of dollars per gram.
Bovine Serum Albumin (BSA) is use as a surrogate to Human Serum Albumin (HSA) in experimental protocols as it presents a practical and scientifically sound alternative. The structural homology of the proteins is evidenced by 76% sequence identity entailing comparable tertiary structures, including similar disulphide bridge arrangements and ligand-binding sites. These structural similarities result in nearly identical spectroscopic properties, with BSA exhibiting comparable UV-visible absorption, fluorescence, and circular dichroism spectra to HSA. This allows BSA to serve as a reliable calibration standard for albumin quantification. further supported by its greater commercial accessibility, cost-effectiveness, and freedom from ethical complications associated with human-derived materials. These combined attributes establish BSA as an optimal surrogate for HSA in spectroscopic albumin detection methodologies.
The following change have been made in the manuscript in the Methods section:
Mid-stream spot urine was collected from healthy volunteers with ethical approval given by Monash university Ethics (2019-21431-34708). All the samples were kept at room temperature until their temperature was stable. The standards albumin (bovine serum albumin: BSA) thereafter herewith referred to as BSA (Mol Wt 66kDa) and Creatinine (CR) 113.12 g/mol was purchased from Merck (St. Louis, MO, U.S.A.). Clinically relevant con-centration ranges were chosen for the BSA and CR.
From one volunteer, we collected four urine samples on different days. From each sample, we prepared 13 fractions to construct spiking models for urine spiked individually with Bovine Serum Albumin (BSA) or creatinine, and 15 fractions for the co-spike (BSA/creatinine) model. Triplicate spectra were recorded for each fraction.
Bovine Serum Albumin (BSA) is widely used as a surrogate for Human Serum Albumin (HSA) in experimental protocols due to its practicality and scientific validity. The two proteins share significant structural homology, with 76% sequence identity, comparable tertiary structures, similar disulphide bridge arrangements, and analogous ligand-binding sites. These similarities result in nearly identical spectroscopic properties, including UV-visible absorption, fluorescence, and circular dichroism spectra, making BSA an effective calibration standard for albumin quantification. Additionally, BSA offers greater commercial availability, cost-efficiency, and avoids the ethical concerns associated with human-derived materials. These attributes collectively establish BSA as an ideal substitute for HSA in spectroscopic methodologies for albumin detection.
2.5. Data analysis and Pre-Processing
After pre-processing, Partial Least Squares Regression (PLS-R) was applied. PLS-R utilizes the relationship between predictors (spectra) and predicted values (concentrations), capturing maximum variables and providing covariance information. PLS-R comprises two matrices: loadings and scores. It is an iterative process that finds vectors best explaining the variance, which are then plotted against each other to provide a linear model. The R² value represents We conducted a 4-fold cross-validation using a Venetian blinds approach, ensuring that no technical replicates were included in the calibration set. As a result, the samples used for testing were not part of the calibration set and were effectively independent of the modelling process.
#Results
Can the authors significantly discuss the clinical relevance of the
creatinine LOD (58.43 mg/L) in co-spiked samples? If ACR is
calculated as albumin (mg/L)/creatinine (g/L), explain whether this
LOD is sufficient for accurate ratio determination in early-stage
DKD?
The significance of achieved LOD is that it is much below the normal physiological range of 0-30 mg/L for albumin and 600-1800 mg/L for creatinine.
As stated above we added the following paragraph
The gold standard to measure albuminuria or proteinuria is the urinary albumin/creatinine ratio (ACR). This ratio corrects the urinary albumin concentration for the urinary creatinine ratio which is dependent on the urinary excretion of creatinine. This is directly determined by renal function and also corrects for different hydration levels. At low creatinine concentrations (near or below the LOD of 58.43 mg/L), the denominator in the ACR calculation becomes less precise due to higher uncertainty or noise in creatinine measurements. This can lead to significant errors or variability in the calculated ACR, particularly for samples near the lower physiological range of creatinine (600–1800 mg/L). The creatinine LOD of 58.43 mg/L is sufficient for most normal physiological ranges but may introduce challenges in accurately calculating the ACR for samples with low creatinine concentrations. This could result in reduced precision or misclassification in certain clinical scenarios, particularly for patients with low urinary creatinine output (creatinine is high in the serum) reflecting reduced renal function. Reducing the LOD for creatinine or enhancing the method's sensitivity would help improve the robustness of ACR measurements at lower concentrations.
We also added the following sentence:
These LoB and LoD calculations provide valuable information about the method's sensitivity and its ability to distinguish between the presence and absence of the analytes in urine samples. They are essential metrics for validating the analytical performance of the developed spectroscopic method for simultaneous quantification of BSA and creatinine in urine.
The significance of the LOD’s achieved as indicated table below are much below the normal physiological range of 0-30 mg/L for albumin and 600-1800 mg/L for creatinine.
#Discussion:
(1) Why didn't the authors Acknowledge the limitations, such as the
use of spiked samples rather than clinical specimens? Can you
address how interferences (e.g., pH, metabolites, medications) in
real urine might affect results?
Though our experiment was based on real human spiked urine samples we consider it advantageous over artificially spiked urine as the real urine better mimics clinical samples. Real human urine samples have all the natural metabolites and other biological components as clinical samples though the difference between both will be the medications present in clinical samples causing drug associated spectral effects leading to pH and different conformational changes in albumin in the sample. We need to be mindful of the limitations of spiked samples as they have known controlled concentrations of purified standards and analytes while clinical samples show natural variation and metabolite ratios and they may not fully assess robustness against biological variability and quite often underestimate matrix interference effects.
(2) Can the authors expand on the pilot trial design (mentioned in
the abstract) to outline how clinical validation will be conducted,
including comparison with ELISA and demographic diversity?
The following changes have been made in this section in the manuscript to address these two points:
Our experiments utilized human spiked urine samples, which we consider advantageous over artificially spiked urine, as real human urine more closely resembles clinical samples. Unlike artificial urine, real human urine contains natural metabolites and biological components similar to those found in clinical specimens. However, clinical samples may also include medications that can introduce drug-associated spectral effects, alter pH levels, and cause conformational changes in albumin, all of which could potentially impact the results.
We acknowledge the limitations of using spiked samples. These samples involve controlled concentrations of purified standards and analytes, which do not fully reflect the natural variability and metabolite ratios present in clinical specimens. Consequently, spiked samples may not adequately assess the robustness of the method against biological variability and can underestimate the effects of matrix interferences. To address these limitations, future studies should incorporate clinical specimens to evaluate the method’s performance under real-world conditions, accounting for the complexities introduced by medications, pH variations, and other biological interferences.
Given the current high cost of laboratory-based albumin/creatinine ratio (ACR) testing using ELISA ($20 per sample), we propose a miniaturised diagnostic device to complement existing testing platforms. This device is designed to enhance accessibility and improve testing efficiency across diverse healthcare settings. While miniaturised devices may not match the analytical precision of conventional large laboratory equipment, their ability to expand diagnostic reach represents a valuable trade-off, particularly in resource-limited contexts.
The spectral data generated by the device can be connected to a cloud-based platform via a smartphone, enabling medical record integration and eHealth applications. Using machine-based algorithms, the data can be securely analysed and transmitted to healthcare providers. The diagnosis is then communicated to both the patient and healthcare professionals, allowing for timely clinical decision-making.
To ensure the method's reliability, validation will involve rigorous testing protocols, including parallel testing with the validated ELISA method through split-sample analysis, blinded testing, inter-laboratory comparisons, and cross-validation with reference laboratories. This comprehensive approach will help establish the accuracy, reproducibility, and robustness of the proposed diagnostic platform.
We have ensured consistent terminology e.g. UV/Visible vs UV-Visible.

Reviewer 2 Report
Comments and Suggestions for Authors
1. The authors claimed a novel approach for direct ultraviolet (UV) region-based quantification of albumin and creatinine in urine. However, established clinical methods (e.g., immunoturbidimetric assays for albumin and enzymatic colorimetric reactions for creatinine) intentionally utilize specific wavelengths (500-600nm) or chromogenic substrates to circumvent interference from urinary background components by using unique molecular interactions (antigen-antibody binding or enzyme-substrate specificity). The authors should explicitly clarify the rationale behind their methodological design, particularly how their UV-based approach addresses the well-documented challenges of spectral overlap (e.g., interference from uric acid, bilirubin, or other proteins) and achieves comparable specificity to gold-standard techniques.
2. Even if the proposed method is theoretically feasible, the experimental evidence presented is inadequate to support its clinical applicability. Critical validations are missing, including: testing on pathological samples (e.g., urine from diabetic kidney disease patients with confirmed albuminuria) ; comparative analysis with established commercial methods; robust statistical evaluation of sensitivity, specificity, and limits of detection (LoD) under real-world matrix effects. Without such data, the method’s diagnostic utility remains unsubstantiated.
3. The introduction overly emphasizes general diabetes epidemiology while inadequately articulating its relevance to the proposed analytical method. A more focused structure is recommended: briefly contextualize diabetic nephropathy as a motivation for urinary biomarker quantification; critically summarize limitations of existing methods (e.g., cost, throughput, or interference issues); clearly state the methodological innovation and its potential advantages. This would strengthen the narrative linkage between clinical need and technical advancement.
4. The data presentation in tables would benefit from reorganization into landscape-oriented tables. This would improve the readability of the manuscript.
Author Response
Reviewer 2
- The authors claimed a novel approach for direct ultraviolet (UV) region-based quantification of albumin and creatinine in urine. However, established clinical methods (e.g., immunoturbidimetric assays for albumin and enzymatic colorimetric reactions for creatinine) intentionally utilize specific wavelengths (500-600 nm) or chromogenic substrates to circumvent interference from urinary background components by using unique molecular interactions (antigen-antibody binding or enzyme-substrate specificity). The authors should explicitly clarify the rationale behind their methodological design, particularly how their UV-based approach addresses the well-documented challenges of spectral overlap
(e.g., interference from uric acid, bilirubin, or other proteins) and achieves comparable specificity to gold-standard techniques.
There are various methods for analysing spectral data, depending on the data type and required information. Simple techniques include peak analysis, such as correlating peak positions with reference spectra, comparing peak intensities, and measuring peak shifts. These can be applied to absolute or second derivative spectra. Peak areas help quantify specific components through integration or curve fitting, enhancing analysis specificity. While single peaks can be analysed, correlating multiple peaks is more reliable due to probability of overlapping bands
A UV-Vis spectroscopy spectrum contains many variables i.e. wavenumber values, which have to be analysed simultaneously. Univariate analysis can be used when investigating a single intensity (peak) band of interest but when we have complex biological mixtures such as urine, there are limitations to the use of univariate analysis and hence warrants a new analytical method i.e. a multivariate approach, which enables the simultaneous analysis of multiple variables at a time and is more often used. Linear and non-linear multivariate regression can be considered the whole spectral range or a selected subsection to generate a linear predictive model from calibration spectra that can be used to quantitate unknown sample data.
The data was divided into calibration and validation sets to ensure robust model development and evaluation. Two urine samples from different days were used for calibration, while one urine sample collected on another day was reserved for validation. For the urine samples spiked individually with BSA and creatinine, the combined calibration and validation sets consisted of 39 fractions. For the co-spiked (BSA/creatinine) experiment, the calibration and validation sets included a total of 45 fractions. Specifically, for urine spiked individually with BSA or creatinine, the calibration set contained 26 fractions, and the validation set contained 13 fractions. For the co-spiked experiment, the calibration set included 30 fractions, while the validation set comprised 15 fractions. Each fraction was measured in triplicate.
The predictive performance of the calibration models was assessed using the independent validation set. The calibration and validation datasets were organised into matrices, denoted as Z and Z_test, respectively. The matrices were structured with dimensions A × B for the calibration set and C × B for the validation set, where A and C represent the number of samples and B corresponds to the number of absorbance wavelengths. The concentrations of BSA or creatinine were subsequently predicted from these datasets, allowing the evaluation of model accuracy and reliability.
From literature we know that the UV-Visible spectroscopy bands of Bilirubin lie between 410 – 440 nm depends on its oxidation state, for our experiment the working region was 200- 320 nm. Yes, true there are other interreferences in urine that will interfere with the spectra.
We acknowledge the limitations of using spiked samples. These samples involve controlled concentrations of purified standards and analytes, which do not fully reflect the natural variability and metabolite ratios present in clinical specimens. Consequently, spiked samples may not adequately assess the robustness of the method against biological variability and can underestimate the effects of matrix interferences. To address these limitations, future studies should incorporate clinical specimens to evaluate the method’s performance under real-world conditions, accounting for the complexities introduced by medications, pH variations, and other biological interferences. We also need to bear in mind that our method is in its infancy and there is plenty or room for further development.
Proposed advantages
Given the current high cost of laboratory-based albumin/creatinine ratio (ACR) testing using ELISA ($20 per sample), we propose a miniaturized diagnostic device to complement existing testing platforms. This device is designed to enhance accessibility and improve testing efficiency across diverse healthcare settings. While miniaturized devices may not match the analytical precision of conventional large laboratory equipment, their ability to expand diagnostic reach represents a valuable trade-off, particularly in resource-limited contexts.
The spectral data generated by the device can be connected to a cloud-based platform via a smartphone, enabling medical record integration and eHealth applications. Using machine-based algorithms, the data can be securely analysed and transmitted to healthcare providers. The diagnosis is then communicated to both the patient and healthcare professionals, allowing for timely clinical decision-making.
To ensure the method's reliability, validation will involve rigorous testing protocols, including parallel testing with the validated ELISA method through split-sample analysis, blinded testing, inter-laboratory comparisons, and cross-validation with reference laboratories. This comprehensive approach will help establish the accuracy, reproducibility, and robustness of the proposed diagnostic platform.
- Even if the proposed method is theoretically feasible, the experimental evidence presented is inadequate to support its clinical applicability. Critical validations are missing, including:
testing on pathological samples (e.g., urine from diabetic kidney disease patients with confirmed albuminuria) ; comparative analysis with established commercial methods; robust statistical evaluation of sensitivity, specificity, and limits of detection (LoD) under real-world matrix effects. Without such data, the method’s diagnostic utility remains unsubstantiated.
Our experiment was based on real human spiked urine samples, which is advantageous over artificially spiked urine as the real urine better mimics clinical samples. Real human urine samples have all the natural metabolites and other biological components as clinical samples though the difference between both will be the medications present in clinical samples causing drug associated spectral effects leading to pH and different conformational changes in albumin in the sample. We need to be mindful of the limitations of spiked samples as they have known controlled concentrations of purified standards and analytes while clinical samples show natural variation and metabolite ratios and they may not fully assess robustness against biological variability and quite often underestimate matrix interference effects.
The following changes have been made in this section in the manuscript to address these two points:
Our experiments utilized human spiked urine samples, which we consider advantageous over artificially spiked urine, as real human urine more closely resembles clinical samples. Unlike artificial urine, real human urine contains natural metabolites and biological components similar to those found in clinical specimens. However, clinical samples may also include medications that can introduce drug-associated spectral effects, alter pH levels, and cause conformational changes in albumin, all of which could potentially impact the results.
We acknowledge the limitations of using spiked samples. These samples involve controlled concentrations of purified standards and analytes, which do not fully reflect the natural variability and metabolite ratios present in clinical specimens. Consequently, spiked samples may not adequately assess the robustness of the method against biological variability and can underestimate the effects of matrix interferences. To address these limitations, future studies should incorporate clinical specimens to evaluate the method’s performance under real-world conditions, accounting for the complexities introduced by medications, pH variations, and other biological interferences.
Given the current high cost of laboratory-based albumin/creatinine ratio (ACR) testing using ELISA ($20 per sample), we propose a miniaturised diagnostic device to complement existing testing platforms. This device is designed to enhance accessibility and improve testing efficiency across diverse healthcare settings. While miniaturised devices may not match the analytical precision of conventional large laboratory equipment, their ability to expand diagnostic reach represents a valuable trade-off, particularly in resource-limited contexts.
The spectral data generated by the device can be connected to a cloud-based platform via a smartphone, enabling medical record integration and eHealth applications. Using machine-based algorithms, the data can be securely analysed and transmitted to healthcare providers. The diagnosis is then communicated to both the patient and healthcare professionals, allowing for timely clinical decision-making.
To ensure the method's reliability, validation will involve rigorous testing protocols, including parallel testing with the validated ELISA method through split-sample analysis, blinded testing, inter-laboratory comparisons, and cross-validation with reference laboratories. This comprehensive approach will help establish the accuracy, reproducibility, and robustness of the proposed diagnostic platform.
- The introduction overly emphasizes general diabetes epidemiology while inadequately articulating its relevance to the proposed analytical method. A more focused structure is recommended: briefly contextualize diabetic nephropathy as a motivation for urinary biomarker quantification; critically summarize limitations of existing methods (e.g., cost, throughput, or interference issues); clearly state the methodological innovation and its potential advantages. This would strengthen the narrative
linkage between clinical need and technical advancement.
Our spectroscopic alternative for point-of-care (POC) testing offers several key advantages: it is high-throughput, fast, cost-effective, and provides real-time results, with sensitivity comparable to the gold standard ACR measurement. Additional benefits include remote testing capabilities, minimizing the need for travel to urban centers, and enabling immediate results in local healthcare settings. The less invasive sampling methods facilitate early detection of conditions, monitoring of treatment effects, and improved patient compliance by reducing loss to follow-up. Furthermore, the ability to perform repeated testing supports better risk stratification and paves the way for personalized medicine.
The following changes have been reflected in the manuscript regarding these two comments:
Urinalysis has been a crucial diagnostic tool since ancient times [20-23]. Currently, dipstick tests are widely used for screening albuminuria due to their simplicity and speed [9,20,24-32]. However, they have low sensitivity (around 70%) for detecting albuminuria, with detection thresholds only reaching 10-20 mg/dL for albumin [56]. These sensitivity constraints in detecting albumin underscore a critical need in healthcare diagnostics for the development of more advanced point-of-care (POC) testing methods to address these limitations, thereby improving diagnostic accuracy and enhancing patient care outcomes.
Given the limitations of current methods, there is a need for cost-effective, non-invasive techniques to measure urinary albumin excretion [36,37]. Spectroscopic approaches, such as UV/Visible spectroscopy, offer promising alternatives for point-of-care testing. This study aims to determine the limit of detection for a UV/Visible spectrometer approach using aqueous urine samples spiked with BSA and creatinine, and to correlate the results with ELISA test values. Our spectroscopic alternative for point-of-care (POC) testing offers numerous advantages: it is high-throughput, fast, provides real-time results, is cost-effective, and demonstrates sensitivity comparable to the gold standard ACR measurement. Additional benefits include remote testing capabilities, reduced need for travel to urban centres, immediate results in local healthcare settings, and less invasive sampling methods. It supports early detection of conditions, effective monitoring of treatment outcomes, and minimizes loss to follow-up. Furthermore, it enables repeated testing, facilitating better risk stratification and advancing personalized medicine.
[56] O. T. Browne and S. Bhandari, Bmj, 2012, 344, e2339.
- The data presentation in tables would benefit from
reorganization into landscape-oriented tables. This would improve
the readability of the manuscript.
We agree but it does utilise more pages so we will leave that decision to the editors.

Reviewer 3 Report
Comments and Suggestions for Authors
My comments are marked with *
- Introduction
* The information in the paragraph on lines 98-105 has already been presented in the paragraph on lines 54-64.
* Paragraph lines 115-120 should be at the end of the introduction
* Please rewrite the introduction in an organized manner.
- Materials and Methods
* How many samples were collected?
2.4. Instrument Parameters and Experimental Procedure
* Where did the authors get this information in lines 216-220?
2.5. Data analysis and Pre-Processing
* How many samples make up the calibration and validation sets?
* How were the calibration and validation sets defined?
* The authors only comment on PLS-R regression, and nothing was described for MatLab.
- Results and Discussion
* Lines 254-268: The authors comment on some things but do not say where the information was taken from. No references are used. This is unacceptable in a scientific paper.
Author Response
Reviewer 3:
- Introduction
* The information in the paragraph on lines 98-105 has already
been presented in the paragraph on lines 54-64.
The following lines have been omitted from lines 54-64 as they have been similarly stated in lines 98-105
Microalbuminuria is characterized by a urinary albumin excretion between 30-300 mg/24hr and macroalbuminuria is characterised by a urinary albumin excretion of greater than 300 mg/24 h. When expressed as urinary ACR (albumin to creatinine ratio), microalbuminuria is characterised by 30-299 mg/g (3-29 mg/mmol) and macroalbuminuria by > 300 mg/g or 30 mg/mmol, in both type 1 and type 2 diabetic patients and other types of chronic kidney disease [1].
* Paragraph lines 115-120 should be at the end of the introduction
Lines 115-120 have been moved to the end of the introduction paragraph
Given the limitations of current methods, there is a need for cost-effective, non-invasive techniques to measure urinary albumin excretion [36,37]. Spectroscopic approaches, such as UV/Visible spectroscopy, offer promising alternatives for point-of-care testing. This study aims to determine the limit of detection for a UV/Visible spectrometer approach using aqueous urine samples spiked with BSA and creatinine, and to correlate the results with ELISA test values.
* Please rewrite the introduction in an organized manner.
The above changes have been made
- Materials and Methods
* How many samples were collected?
The following changes have been made in the manuscript to address this point
From one volunteer, we collected four urine samples on different days. From each sample, we prepared 13 fractions to construct spiking models for urine spiked individually with Bovine Serum Albumin (BSA) or creatinine, and 15 fractions for the co-spike (BSA/creatinine) model. Triplicate spectra were recorded for each fraction.
2.4. Instrument Parameters and Experimental Procedure
* Where did the authors get this information in lines 216-220?
We have added the following reference: Technical note
[49] Loughrey, S.; Mannion, J.; Matlock, B. Nucleic acid measurements at 260 nm; 2017, Technical notes (TN52918) www.nanodrop.com
2.5. Data analysis and Pre-Processing
* How many samples make up the calibration and validation sets?
The following text was added to the paragraph below:
For the urine spiked individually with BSA and Creatinine, the calibration and validation sets together comprised 39 fractions. In contrast, the co-spiked (BSA/Creatinine) experiment included 45 fractions for calibration and validation.
* How were the calibration and validation sets defined?
The following text was added to the paragraph below:
For urine spiked individually with BSA and Creatinine, the calibration sets consisted of 26 fractions, while the validation sets included 13 fractions. In the co-spiked (BSA/Creatinine) experiment, the calibration sets comprised 30 fractions, and the validation sets included 15 fractions, with spectra for each fraction recorded in triplicate.
* The authors only comment on PLS-R regression, and nothing
was described for MatLab.
The PLS-regression analysis was conducted using the MatLab PLS-Toolbox software package (version 8.8.1).
PLS Toolbox is a specialized MATLAB add-on designed for advanced chemometric analysis. It provides an array of tools for multivariate chemometric analysis: Partial Least Squares (PLS) regression modelling, spectral processing, data pre-processing and exploration and model validation and optimization.
The following change was made in the manuscript for this query are reflected below:
Chemometric analysis was employed to determine the concentrations of BSA and creatinine from the spectral data following appropriate spectral pre-processing. Regression models were developed using MATLAB (MathWorks, Natick, MA, USA) in combination with the PLS Toolbox (Eigenvector Research, Manson, WA, USA). The PLS Toolbox, a specialized MATLAB add-on, provides an extensive suite of tools for advanced chemometric analysis, including Partial Least Squares (PLS) regression modelling, spectral processing, data pre-processing, exploratory analysis, and model validation and optimization.
The data was divided into calibration and validation sets to ensure robust model development and evaluation. Two urine samples from different days were used for calibration, while one urine sample collected on another day was reserved for validation. For the urine samples spiked individually with BSA and creatinine, the combined calibration and validation sets consisted of 39 fractions. For the co-spiked (BSA/creatinine) experiment, the calibration and validation sets included a total of 45 fractions. Specifically, for urine spiked individually with BSA or creatinine, the calibration set contained 26 fractions, and the validation set contained 13 fractions. For the co-spiked experiment, the calibration set included 30 fractions, while the validation set comprised 15 fractions. Each fraction was measured in triplicate.
The predictive performance of the calibration models was assessed using the independent validation set. The calibration and validation datasets were organised into matrices, denoted as Z and Z_test, respectively. The matrices were structured with dimensions A × B for the calibration set and C × B for the validation set, where A and C represent the number of samples and B corresponds to the number of absorbance wavelengths. The concentrations of BSA or creatinine were subsequently predicted from these datasets, allowing the evaluation of model accuracy and reliability.
- Results and Discussion
Lines 254-268: The authors comment on some things but do not
say where the information was taken from. No references are
used. This is unacceptable in a scientific paper.
We have added the following 6 references to this section of the paper.
Figure 1 presents the UV/Visible absorption spectra of Bovine Serum Albumin (BSA) and creatinine in water at a concentration of 150 mg/L. The spectrum for BSA features a prominent peak at 217 nm, which corresponds to the peptide (amide) chromophore of the protein, primarily associated with the side chains of tryptophan (W) and phenylalanine (F). Additionally, a peak at 280 nm arises from the combined contributions of the main light-absorbing amino acids: tryptophan (W), tyrosine (Y), and phenylalanine (F). Although other light-absorbing amino acids are present in proteins, their absorption intensities in this spectral range are less pronounced compared to W, Y, and F. [50, 51, 52, 53, 54, 55]
References:
Text format:
[50] Demchenko, A. P. Ultraviolet spectroscopy of proteins; Springer, 1986. DOI: 10.1007/978-3-642-70847-3.
[51] Nixon, A. E. Therapeutic Peptides Methods and Protocols Preface. In THERAPEUTIC PEPTIDES: METHODS AND PROTOCOLS, Nixon, A. E. Ed.; Vol. 1088; 2014; pp V-VI.
[52]Donovan, J. W. Changes in Ultraviolet Absorption Produced by Alteration of Protein Conformation. Journal of Biological Chemistry 1969, 244 (8), 1961-1967. DOI: https://doi.org/10.1016/S0021-9258(18)94353-X.
[53] Glazer, A.; Smith, E. Effect of Denaturation on the Ultraviolet Absorption Spectra of Proteins. Journal of Biological Chemistry 1960, 235, PC43-PC44. DOI: 10.1016/S0021-9258(18)64584-3.
[54] Lin, T. J.; Yen, K. T.; Chen, C. F.; Yan, S. T.; Su, K. W.; Chiang, Y. L. Label-Free Uric Acid Estimation of Spot Urine Using Portable Device Based on UV Spectrophotometry. Sensors 2022, 22 (8). DOI: 10.3390/s22083009.
[55] Hammond, B. R.; Johnson, B. A.; George, E. R. Oxidative photodegradation of ocular tissues: Beneficial effects of filtering and exogenous antioxidants. Experimental Eye Research 2014, 129, 135-150. DOI: 10.1016/j.exer.2014.09.005.

Round 2
Reviewer 1 Report
Comments and Suggestions for Authors
Thank you for addressing all relevant questions and comments raised during the first stage of peer-review.
Author Response
No changes required
Reviewer 2 Report
Comments and Suggestions for Authors
The author has revised the manuscript and answered several questions, though I have few more questions:
- Please clarify the number of healthy volunteers whose urine samples were analyzed in this study. Were they similar in the UV region? The spectral data of normal urine presented in Figure 1 are unclear to me—is the presented spectrum typical?
- Due to interindividual variability in diet, metabolism, and physiology, healthy human urine may contain numerous metabolites with overlapping absorption features close to the region used by the author for PLS-R. For instance, vitamin B2 and vitamin C, when consumed in excess, are excreted in urine and exhibit characteristic absorption peaks near 250 nm. Has the author accounted these potential interferences in the analysis?
- The reliability of this method should be evaluated by directly testing patient-derived urine samples against established techniques. The author did mention this in the revised manuscript that such experiment will be conducted in the future. What practical challenges prevented the authors from conducting such comparisons currently?
Author Response
See attached word document

Reviewer 3 Report
Comments and Suggestions for Authors
After correction according to the reviewers' suggestions (highlighted in the text), the manuscript is of better quality, with more informative details, and is ready for publication.
Author Response
No changes required